# Exploring the influence of women's leadership and corporate governance on operational liquidity: The glass cliff effect

Dongli Cao[1]*, Safdar Husain Tahir[2]*, Syed Maisam Raza Rizvi[3], Khuda Bakhsh Khan[4]

1 School of Law, Southwestern University of Finance and Economics, Chengdu, Sichuan China, 2 Lyallpur Business School, Government College University, Faisalabad, Pakistan, 3 Department of Accountancy and Finance, University of Otago, Dunedin, New Zealand, 4 Department of Education, Government College University Faisalabad, Faisalabad, Pakistan

* 1200201z1001@smail.swufe.edu.cn (DC); drsafdargcuf@gmail.com (SHT)

**Data Availability Statement:** All relevant data are within the paper and its Supporting Information files.

## Abstract

This research investigates the glass cliff effect and the positions held by women in leadership roles, focusing on their impact on operational liquidity. The study delves into the relationship between corporate governance attributes and operational liquidity in 60 non-financial companies listed on the Pakistan Stock Exchange during Covid-19. Utilizing Quine-McCluskey technique and fuzzy set Qualitative Comparative Analysis (fsQCA), it examines the combined effect of Women on the Board, Board Size, Ownership by Blockholders, Board Qualifications and Busy Directors on Operational Liquidity. The necessary condition analysis (NCA) emphasises that firms can operate without reliance on any particular variable taken in the study. The sufficiency analysis provided an expanded understanding of the three conditions leading to the same outcome both before and during the pandemic. This research highlights the significance of the glass cliff effect and emphasizes the pivotal role of women in effectively managing liquidity during times of crisis. Additionally, it provides valuable insights for policymakers regarding the impact of Covid-19 on the interplay between corporate governance characteristics and operational liquidity.

## Introduction

The biggest challenge facing management in the non-financial industry is effectively managing operational liquidity, particularly in challenging economic times like Covid-19 [1]. Research scholars have long been focused on operating liquidity, which garnered significant attention during the Covid-19 pandemic [2–4], as it caused massive liquidity problems for the non-financial sector, thereby impacting companies' profits globally in 2020–21 [5]. This, in turn, led to liquidity risk management challenges for numerous companies, particularly those in the non-financial sector, creating a surge in market demand for liquidity. The financial industry also faced immense stress and a severe threat of default during the pandemic [6], ultimately impacting the operating liquidity of the non-financial sector. However, effective corporate practices and board attributes including presence of women on board enhance firms' liquidity [7].

**Funding:** The author(s) received no specific funding for this work.

Effective management through good corporate governance of a firm's operating liquidity is crucial as it involves maintaining a balance between current assets and liabilities. This balance ensures the firm can cover any potential cash shortages during financial difficulties in Covid-19 and seize lucrative investment openings [8–10]. However, excessive liquidity can negatively impact the firm's profitability [11], as holding more liquid assets means that a more significant portion of the firm's support is not generating returns [12], resulting in lower income and tax disadvantages that can lower overall profitability [13]. Moreover, the ease of misappropriation of liquid assets by opportunistic managers can further exacerbate this problem, as these resources can be used for personal gain instead of financing the business's growth opportunities [14–16].

Prior research has identified a number of firm-level characteristics that affect a company's operating liquidity. These characteristics include both internal and external factors, including the institutional environment, investor protection, and the effectiveness of the economic system [17, 18]. Internal factors include business size, financial limits, credit ratings, and growth potential [9]. In this research, we intend to examine the combined effect of corporate governance attributes such as women on the board, the board size, ownership of block holders, director's qualification, and busy directors on operating liquidity before and during covid-19 under the glass cliff phenomenon.

Several reasons led to the exploration of how corporate governance and women's leadership affect operational liquidity in the context of the Glass Cliff effect. First, the necessity for efficient liquidity risk management was brought to light by the ongoing significance of operational liquidity in research, which was further emphasised by the difficulties non-financial firms had during the Covid-19 pandemic. Second, the pandemic's effects on the liquidity of non-financial sectors and the stress experienced by the banking industry brought to light the critical role that corporate practices and board composition play in a firm's ability to sustain liquidity. It becomes crucial for good corporate governance, especially in the area of operating liquidity management, to strike a balance between current assets and commitments during crisis times.

Third, the "Glass Cliff effect" is a fundamental theoretical framework used in this study, which suggests that women are frequently assigned to leadership positions in times of crisis or uncertainty, when the likelihood of failure is increased [19–21]. It was believed that women's natural abilities, such as risk awareness, empathy, and teamwork, would help them overcome hitherto unheard-of challenges [22, 23]. However, taking on leadership roles during times of crisis may also bring extra challenges and prejudices for women and minorities, which may explain why they are underrepresented in such jobs. Business, politics, and sports have all been seen to be affected by this phenomenon [24, 25].

Last but not least, in addition to examining the representation of women on boards, the study looks into other characteristics of boards, including size, ownership of blockholders, director's qualification, and busyness. A thorough investigation of the complex interactions of gender dynamics, corporate governance, and operational liquidity is made possible by this dual theoretical approach. This theoretical framework directs the meticulous variable selection and overall research design, guaranteeing an in-depth analysis of the complexity surrounding the Glass Cliff effect in the particular context of path way configurational analysis.

The aim of this research is to enhance to the body of knowledge already existing on the glass cliff effect, particularly in the context of crisis management during the Covid-19 pandemic [26–29]. The focus would be on countries with poor governance systems, like Pakistan [30, 31], where women and minorities may face additional barriers and biases when taking on leadership roles during times of crisis. Specifically, we seek to answer the question of whether board attributes, including women on board, play a substantial role in shaping a firm's operating liquidity in the context of the class cliff effect.

### Research question

How do board attributes, including women on board, impact operating liquidity before and during Covid-19 crisis under the glass cliff phenomenon in nations with weak governance institutions, such as Pakistan?

### Sub-research questions

- How does the association between operating liquidity and board size change before and during the Covid crisis?

- How does block holder ownership impact operating liquidity prior to and during the Covid-19 crisis?

- How does operating liquidity prior to and following the Covid-19 crisis depend on the directors' qualifications?

- Do busy directors have an impact on operating liquidity before and after the Covid-19 crisis?

- How does the glass cliff effect affect the relationship between operating liquidity before and after the Covid-19 crisis and the presence of women on boards?

### Research objectives

- To assess how the glass cliff effect may affect this relationship and the role that women on the board played in deciding operating liquidity before and during Covid crisis.

- To examine at the overall effect of board characteristics on operating liquidity both before and during the Covid crisis in countries like Pakistan with weak governance structures.

- To investigate how board size affected operating liquidity before and during the Covid crisis.

- To look into how stockholder ownership affected operating liquidity prior to and during the Covid crisis.

- To investigate how director qualifications affected operating liquidity both before and during the Covid crisis.

### Literature review

Several studies have found a strong relationship between outstanding corporate governance characteristics and high financial and operational efficiency levels, data reporting quality, and firm liquidity, with minimal information variability [32, 33]. Improved governance mechanisms, precisely board attributes, have been shown to increase information visibility, leading to managers' increased disclosure of internal financial data [34], ultimately reducing cost and increasing financial performance [35]. This study adds to the body of knowledge on company performance and corporate governance in two ways. In the first place, it particularly looks at how women's contributions to corporate boards have affected crisis management—like it did during the COVID-19 pandemic. Second, it clarifies the channels through which different board attributes alone or collectively affect overall company success in the context of glass cliff effect.

## Women on board and operating liquidity

The participation of women on corporate boards has been shown to enhance governance by increasing efficiency and monitoring responsibilities [36, 37]. This is because female directors are more vigilant than male directors, increasing the board's monitoring responsibility [38]. On the other hand, market competitiveness determines the influence of women's directors on a board's efficacy since female directors are more careful in competitive conditions, slowing decision-making and weakening governance [39]. It may negatively impact operating liquidity [40]. Ahmed and Ali [41] researched the influence of women on board at Australian companies and discovered a constructive relationship.

Pakistani culture is well-known because of its diverse corporate environment and predominantly male society [42], it is interesting to examine the association amid women on board and corporate liquidity, particularly in the context of glass cliff effect. The majority of enterprises in Pakistan are also family-run [43, 44], where women are likely to become directors of their companies. Furthermore, women directors usually bring new resources to the firm, which can help the business's public image and market reputation [45]. As a consequence, the nomination of female directors is likely to produce a favourable reaction from investors, perhaps improving the financial health of a firm [37]. In this study, it would be interesting to examine how a company's performance is impacted by the composition of the board, especially with regard to the representation of women. The modern research methodology, fsQCA, makes it possible to find several roots, which lead to performance in the context of the COVID-19 pandemic. Thus, the first proposition of the study is proposed below:

$P_1$: A higher representation of women on corporate boards is anticipated to play a significant role in effectively managing liquidity, especially in the context of the COVID-19 pandemic.

## Board size and operating liquidity

Past research has discovered the association between board size and the allocation of liquid assets for the effective operation of a firm. A large board, measured by the number of directors, is often considered more advantageous for decision-making regarding operating liquidity [46, 47], particularly in times of crisis [48]. Anderson, Mansi [49] indicate a colossal board can also enhance corporate governance mechanisms and risk management strategies, particularly in times of uncertainty and information asymmetry. However, some studies have found non-significant connection between board size and managing operating liquidity, particularly during crises [50, 51]. The study aims to contribute to the literature on corporate governance by employing a novel research methodology, fsQCA, to explore the joint effects of board size on output. Thus, the second proposition is proposed as below:

$P_2$: A larger board size is expected to play a more active role in shaping and managing the liquidity of a firm, particularly in the context of the COVID-19 pandemic.

## Ownership of block-holders and operating liquidity

Previous research has yielded conflicting findings regarding the relationship between block-holder ownership and operating liquidity. While Ding, Shen [52] report a negative association, Sarker and Fisher [53] demonstrate that block-holder ownership can increase liquidity by allowing for more effective monitoring and access to insider knowledge. This study aims to look into how block-holder ownership affected operating liquidity before and after the COVID-19 outbreak. Al-Rassas [54] suggests that information asymmetry may be a key factor, as stockholders may have access to privileged information not available to other shareholders. This can result in poor liquidity, particularly in businesses with limited shareholders and

substantial block-holder ownership [55]. Block holders' superior information may also exacerbate adverse selection issues [56]. Thus, the third proposition is proposed as follows:

$P_3$: The operating liquidity of a firm during the COVID-19 pandemic is expected to benefit significantly from a higher level of blockholders ownership.

## Busy director and operating liquidity

Busy directors, who hold multiple directorships, are considered valuable assets to the board due to their monitoring abilities, advisory skills, and networking contacts [57–60]. Masulis and Mobbs [57] argue that busy directors can bring external expertise and information to improve company performance, such as in major decisions like acquisitions. Therefore, firms, who have more busy directors on board are expected to have better operating liquidity. However, there is a risk that busy directors may overextend themselves, spending less time on each board, and failing to fulfil their obligations, negatively impacting performance [61, 62]. Cooper and Uzun [63] suggest that busy boards negatively affect firm liquidity as their monitoring and advisory capacity declines. In contrast to traditional regression methods, our research employs an innovative approach, fsQCA, to examine the collective influence of factors such as busy directors alongside the presence of women on the board, board size, blockholders ownership, and director qualifications on operational liquidity during times of crisis. This contributes to a comprehensive understanding from multiple perspectives. Thus, next proposition is framed as below:

$P_4$: The presence/absence of a higher number of busy directors is expected to contribute to increase / decrease level of operating liquidity in firms during crisis times.

**Director's qualification and operating liquidity.** The study explores the impact of directors' qualifications on operating liquidity. Higher education is associated with improved cognitive abilities, skill sets, and risk-taking mind-sets. Qualified directors are better equipped to comprehend and evaluate information, analyze the consequences of decisions, and develop solutions to complex problems [64–66]. Studies have shown that firms with highly qualified and prominent boards of directors have better operating liquidity and lower costs of external funding [67, 68].

However, this may also result in these firms needing more capital. Therefore, the director's qualifications are negatively linked to liquidity [69–72]. Our research assesses the synergistic impact of director qualifications in conjunction with the presence or absence of busy directors, the inclusion of women on the board, board size, and blockholders ownership on operational liquidity both prior to and following the onset of the pandemic crisis. Based on the insights derived from this examination, we suggest the following proposition:

$P_5$: The higher level of professional qualifications among directors is expected to have a notable influence on operating liquidity both prior to and following the pandemic crisis.

## Conceptual framework

Fig 1 displays the study's theoretical framework, which examines the complex relationship between operational liquidity and corporate governance issues in the context of the glass cliff phenomena, especially in times of crisis such as the COVID-19 epidemic. Past literature repeated shows direct relationship between corporate governance and operational liquidity [32, 33]. By concentrating on two important variables within the context of the glass cliff, our study adds to this body of information. First, we examine how women's contributions to corporate boards have affected crisis management, acknowledging their part in improving accountability, efficiency, and governance.

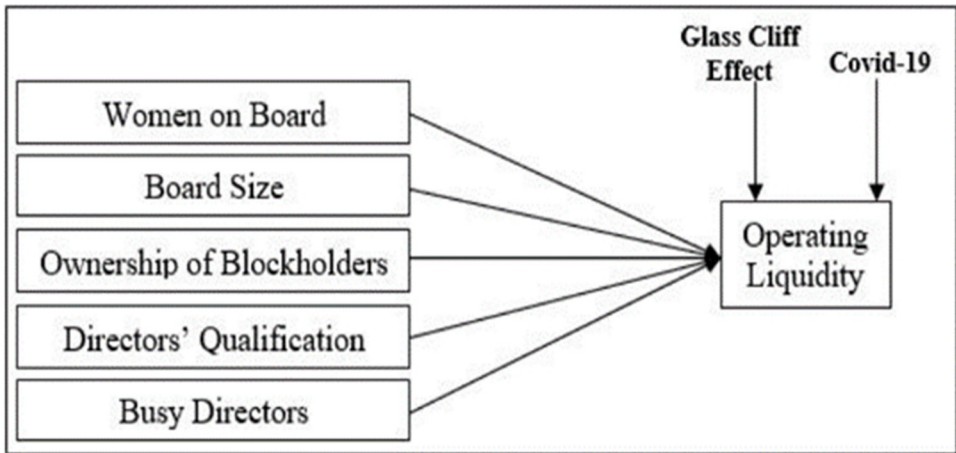

**Fig 1. Conceptual framework.**

Second, taking into account the phenomena of glass cliff effect, we investigate how various board characteristics, such as board size, blockholders ownership, busy directors, and directors' qualification, both individually and jointly impact overall firm success. We hope to shed light on the complex interactions between these variables by using the cutting-edge research approach fsQCA. By doing so, we will be able to better comprehend the dynamics of corporate performance in the face of novel problems. The aforementioned ideas provide a methodical framework for investigating the complex dynamics present in the glass cliff scenario, illuminating the various mechanisms by which corporate governance impacts operational liquidity in times of crisis.

## Method

### Configurational approach

The study used the fsQCA technique, which comprises a discussion between theories and actual data utilising a moderate number of cases, often between 15 and 65 [73, 74]. We have compiled data from 60 non-financial companies that are publicly traded on the Pakistan Stock Exchange (PSX) during the years 2018 to 2021. Our objective is to investigate the relationship between operational liquidity and corporate governance attributes, taking into account the impact of the Covid-19 pandemic and the phenomenon known as the "glass cliff." To assess the changes in the corporate landscape before and after the onset of the Covid-19 pandemic, we have conducted an analysis of the years 2018 and 2021. The key details of our sample are presented in Table 1 for reference.

**Table 1. Characteristics of the sample.**

| Input | 2018 | | | | 2021 | | | |
|---|---|---|---|---|---|---|---|---|
| | Mean | Std. Dev | Min | Max | Mean | Std. Dev | Min | Max |
| WOB | 0.07 | 0.08 | 0 | 0.33 | 0.1 | 0.07 | 0 | 0.33 |
| BS | 8.55 | 1.77 | 7 | 14 | 8.56 | 1.67 | 7 | 13 |
| OBH | 0.45 | 0.25 | 0 | 0.93 | 0.47 | 0.25 | 0 | 0.93 |
| DQ | 0.56 | 0.21 | 0.11 | 1 | 0.57 | 0.23 | 0 | 1 |
| BD | 0.71 | 0.21 | 0 | 1 | 0.71 | 0.23 | 0 | 1 |
| CR | 1.53 | 1.07 | 0.11 | 5.64 | 1.42 | 0.75 | 0.07 | 4.63 |

Table 1 lists the characteristics of the sample for six variables: the number of women on the board (WOB), the size of the board (BS), the ownership of block holders (OBH), the board's qualifications (DQ), the number of busy directors (BD), and the current ratio (CR) before (2018) and after (2021) the Covid-19 pandemic. The mean, standard deviation, lowest and maximum values for both years are shown in the table. The mean value of WOB increased from 0.07 to 0.1 while the mean value of CR decreased from 1.53 to 1.42, showing changes in the sample characteristics. The disparities in standard deviations range from 0.0 to 0.08, while the differences in means range from 0.01 to 0.04. These results emphasise the significance of comparing the variable configurations prior to and following the pandemic in the context of the glass cliff effect. The data is calibrated on the 95th, 50th, and 5th percentiles to conduct necessary and adequate condition assessments using fuzzy set qualitative comparative analysis (fsQCA) [75–80].

## Necessary conditions analysis

Table 2 displays the results of the necessary condition analysis using fsQCA (fuzzy-set Qualitative Comparative Analysis) before and after Covid-19. The table displays consistency and coverage values for each input and its absence for observing necessary condition analysis (NCA) as used by Rekik and Bergeron [81] and Roig-Tierno, Gonzalez-Cruz [82].

Looking at Table 2, there are some differences in the values before and after Covid-19. For example, the consistency of WOB decreases from 0.92 to 0.71, and the consistency of BD increases from 0.62 to 0.68. Overall, it appears that there are no necessary conditions of any individual input variable before or after Covid-19 as none of the consistency and coverage values simultaneously reach the threshold of 0.90, which is typically considered the minimum acceptable level for necessary conditions of individual input variable in fsQCA [83, 84].

## Sufficient conditions analysis

The study's next step is sufficiency condition analysis before and after covid-19. We must compare the PRI consistency scores with the raw consistency scores to evaluate the truth in Table 3 regarding proportional reduction in inconsistency (PRI). The PRI consistency score gauges how much a particular configuration lessens the truth table's overall inconsistency. The higher the PRI consistency score, the more influential the configuration reduces inconsistency.

**Table 2. Necessary conditions analysis before and after Covid-19.**

| Input | 2018 | | | | 2021 | | | |
|---|---|---|---|---|---|---|---|---|
| | Presence | | Absence | | Presence | | Absence | |
| | Con (%) | Cov (%) | Con (%) | Cov (%) | Con (%) | Cov (%) | Con (%) | Cov (%) |
| WOB | 92 | 63 | 89 | 73 | 71 | 77 | 58 | 74 |
| ~WOB | 61 | 83 | 89 | 56 | 76 | 61 | 82 | 76 |
| BS | 56 | 59 | 62 | 78 | 54 | 54 | 67 | 77 |
| ~ BS | 79 | 64 | 68 | 65 | 77 | 67 | 60 | 60 |
| OBH | 65 | 70 | 67 | 74 | 74 | 69 | 62 | 67 |
| ~OBH | 72 | 64 | 68 | 73 | 64 | 59 | 71 | 76 |
| DQ | 65 | 61 | 72 | 80 | 70 | 59 | 73 | 72 |
| ~ DQ | 79 | 70 | 65 | 69 | 67 | 69 | 59 | 69 |
| BD | 62 | 59 | 65 | 74 | 68 | 57 | 74 | 72 |
| ~ BD | 72 | 64 | 63 | 66 | 67 | 69 | 56 | 67 |

The symbol (~) shows the absence of a condition; Con.: Consistency; Cov.: Coverage. All values in percentage

**Table 3. Truth table 2018.**

| WOB | BS | OBH | DQ | BD | f | CR | Raw-consist | PRI-consist | SYM-consist |
|---|---|---|---|---|---|---|---|---|---|
| 1 | 0 | 1 | 1 | 1 | 1 | 1 | 0.922 | 0.500 | 0.531 |
| 1 | 0 | 1 | 0 | 0 | 1 | 1 | 0.892 | 0.543 | 0.604 |
| 1 | 0 | 0 | 0 | 0 | 1 | 1 | 0.883 | 0.476 | 0.502 |
| 1 | 0 | 0 | 1 | 1 | 1 | 1 | 0.872 | 0.265 | 0.265 |
| 1 | 0 | 1 | 1 | 0 | 3 | 1 | 0.851 | 0.448 | 0.481 |
| 1 | 1 | 0 | 0 | 0 | 3 | 0 | 0.846 | 0.491 | 0.499 |
| 1 | 0 | 0 | 1 | 0 | 1 | 0 | 0.840 | 0.260 | 0.260 |
| 1 | 1 | 1 | 1 | 0 | 1 | 0 | 0.837 | 0.290 | 0.290 |
| 1 | 1 | 0 | 1 | 1 | 3 | 0 | 0.787 | 0.068 | 0.070 |
| 1 | 1 | 1 | 1 | 1 | 2 | 0 | 0.752 | 0.030 | 0.030 |

Table 3 shows that the PRI consistency scores apan from 0.068 to 0.543, which are all below the raw consistency scores, spaning from 0.752 to 0.922.

Similarly, Table 4 shows that the PRI consistency scores range from 0.097 to 0.793, while the raw consistency scores range from 0.760 to 0.956. The PRI consistency scores are generally lower than the raw consistency scores, indicating some inconsistency not accounted for by the configurations in the table. However, some configurations have higher PRI consistency scores than others, suggesting they are more effective in reducing overall inconsistency [85]. Therefore, based on these findings, we can proceed with the sufficiency analysis in the study.

Table 5 presents the results of sufficiency analysis for the model CR = f(WOB, BS, OBH, DQ, BD) using fsQCA most parsimonious specify analysis with Quine-McCluskey algorithm for two time periods: 2018 (pre-Covid-19) and 2021 (post-Covid-19). A frequency cutoff of 1 and a consistency cutoff of 0.85 were used in both analyses. In 2018, three combinations of

**Table 4. Truth table 2021.**

| WOB | BS | OBH | DQ | BD | f | CR | Raw-consist | PRI-consist | SYM-consist |
|---|---|---|---|---|---|---|---|---|---|
| 1 | 0 | 1 | 1 | 1 | 1 | 1 | 0.956 | 0.776 | 0.780 |
| 0 | 1 | 1 | 0 | 1 | 2 | 1 | 0.946 | 0.452 | 0.465 |
| 1 | 0 | 1 | 1 | 0 | 2 | 1 | 0.946 | 0.793 | 0.804 |
| 1 | 0 | 1 | 0 | 0 | 1 | 1 | 0.916 | 0.583 | 0.583 |
| 1 | 1 | 1 | 1 | 0 | 1 | 1 | 0.897 | 0.515 | 0.515 |
| 0 | 0 | 1 | 0 | 1 | 2 | 1 | 0.896 | 0.566 | 0.620 |
| 1 | 1 | 1 | 0 | 0 | 1 | 1 | 0.892 | 0.291 | 0.291 |
| 1 | 0 | 0 | 0 | 0 | 1 | 1 | 0.868 | 0.478 | 0.478 |
| 0 | 1 | 1 | 0 | 0 | 2 | 1 | 0.867 | 0.327 | 0.361 |
| 1 | 0 | 0 | 0 | 1 | 3 | 1 | 0.861 | 0.339 | 0.339 |
| 1 | 0 | 0 | 1 | 0 | 1 | 1 | 0.859 | 0.289 | 0.289 |
| 1 | 1 | 0 | 1 | 1 | 1 | 1 | 0.858 | 0.224 | 0.239 |
| 0 | 1 | 0 | 0 | 1 | 1 | 0 | 0.831 | 0.156 | 0.156 |
| 0 | 0 | 0 | 0 | 1 | 1 | 0 | 0.810 | 0.294 | 0.294 |
| 0 | 1 | 0 | 1 | 0 | 2 | 0 | 0.798 | 0.219 | 0.227 |
| 0 | 1 | 0 | 0 | 0 | 2 | 0 | 0.789 | 0.345 | 0.345 |
| 1 | 1 | 0 | 0 | 0 | 1 | 0 | 0.788 | 0.311 | 0.314 |
| 0 | 1 | 0 | 1 | 1 | 6 | 0 | 0.771 | 0.097 | 0.098 |
| 0 | 1 | 1 | 1 | 1 | 6 | 0 | 0.760 | 0.115 | 0.121 |

**Table 5. Sufficiency analysis.**

| 2018 | | | | 2021 | | | |
|---|---|---|---|---|---|---|---|
| Path | Raw-Cov | Unique-Cov | Consist | Path | Raw-Cov | Unique-Cov | Consist |
| ~BS*~DQ | 0.631 | 0.029 | 0.805 | WOB*~BS | 0.596 | 0.059 | 0.841 |
| ~BS*OBH | 0.593 | 0.050 | 0.767 | OBH*~DQ | 0.488 | 0.118 | 0.794 |
| ~BS*BD | 0.516 | 0.023 | 0.762 | WOB*DQ | 0.562 | 0.059 | 0.829 |
| Solution coverage | 0.756 | | | Solution coverage | 0.796 | | |
| Solution consistency | 0.813 | | | Solution consistency | 0.853 | | |

conditions were found to be sufficient for the outcome of interest, i.e., operating liquidity. These combinations are: ~BS*~DQ, ~BS*OBH, and ~BS*BD. The solution coverage, the percentage of cases covered by all solution paths combined, is 0.756. This means that the solutions explain 75.6% of the cases in the truth table. The solution consistency (0.813) indicates that the solutions are relatively consistent in explaining the outcomes in the truth table. Overall, the sufficiency analysis shows that the discovered solutions fit the data well because they account for a sizable fraction of the cases and are generally consistent.

For 2021, however, the research found three sets of circumstances that were adequate to produce the desired result. These had solution consistency of 0.853 and solution coverage of 0.796 respectively. They were WOB*DQ, OBH*DQ, and WOB*BS. The consistency of the three conditions reported in 2018 ranges from 0.762 to 0.805, which is substantially less consistent than the three conditions found in 2021. The combination of conditions discovered in 2021, on the other hand, has more consistency, ranging from 0.829 to 0.841. The proportion of cases with the desired outcome, known as the solution coverage, was consistent during both time periods. These results imply that the Covid-19 epidemic had a significant impact on operational liquidity.

The operating liquidity is the outcome of interest, and the fsQCA study results for 2018 and 2021 reveal differences in the recognised sufficient combinations of criteria or variables. Conditions such BS*DQ, BS*OBH, and BS*BD that are associated to the absence of a woman in a corporate leadership role were recognised varieties in 2018. The 2021 results, however, demonstrate that the combinations of WOB*BS and WOB*DQ were determined to be sufficient for the outcome of interest, demonstrating that the presence of a woman in a position of corporate leadership is linked to higher operating liquidity. This result could suggest a shift from the glass cliff effect, as companies may be more willing to appoint women to leadership positions when facing crises. These findings are in line with the "glass cliff," which contends that women are more likely to be given leadership roles in crisis situations or troubled businesses [86]. Thus, the study proves the glass cliff effect of the COVID-19 pandemic and indicates that "*A higher representation of women on corporate boards is anticipated to play a significant role in effectively managing liquidity, especially in the context of the COVID-19 pandemic*". The current study is in line with the existing studies in literature in the context of glass cliff [20, 24, 87, 88].

According to the study, no evidence suggests that a larger board size plays a role in managing firm liquidity amidst the COVID-19 pandemic. The analysis of the fsQCA data shows that before COVID-19, the role of BS in managing CR was represented by three paths: ~BS*~DQ, ~BS*OBH, and ~BS*BD. The path with the highest raw coverage was ~BS~DQ, but its low unique coverage suggests it could have been more informative. The path with the highest unique coverage was ~BS*OBH, indicating that this combination of variables was more helpful in identifying the solution. The path ~BS*BD had the lowest raw and unique coverage, suggesting it needed to be more informative. After COVID-19, the role of BS in managing CR was captured by the path WOB*~BS, which had a lower raw coverage than the BS*DQ path before

COVID-19. However, its higher unique coverage suggests that this combination of variables was more helpful in identifying the solution. The consistency score of 0.841 indicates that the relationship between WOB and ~BS was stable across cases after COVID-19. However, negative signs indicate the irrelevancy of BS. The study's findings suggest that *board size was irrelevant in managing firm liquidity before and after the COVID-19 pandemic*. Our study is in line with the study of Mulcahy and Linehan [88].

Based on Table 5, the role of OBH before Covid-19 in managing CR for fsQCA analysis was captured by the path ~BS*OBH. This path had a raw coverage of 0.593, indicating that this combination of variables was present in the data. Its unique coverage of 0.050 suggests that this combination was informative or valuable in identifying the solution. The combination of BS and OBH was rather consistent across cases before Covid-19, according to the consistency score of 0.767.

However, Table 5 depicts the course OBH*DQ took following Covid-19 in terms of OBH's responsibility for operating liquidity management. In contrast, prior to COVID-19, the path BS*OBH had a raw coverage of 0.593, a unique coverage of 0.050, and a consistency score of 0.767, capturing the importance of OBH in managing operating liquidity for fsQCA analysis. This implies that prior to COVID-19, the combination of BS and OBH was likewise fairly steady and instructive. Before and after COVID-19, OBH had a comparable role in maintaining operating liquidity for fsQCA analysis, but the precise paths that recorded this relationship were different. Before COVID-19, BS*OBH was an absolute path, but after COVID-19, OBH*DQ became the more crucial way. OBH so significantly contributes to operating liquidity both before and after *Covid-19*. this study is in line with search conducted by Garel and Petit-Romec [89] and Zhang, Gao [90].

Similarly, the path ~BS*BD played a role in managing operating liquidity for fsQCA analysis before COVID-19. The path had a raw coverage of 0.516, indicating that the combination of ~BS and BD was significant for operating liquidity. The low unique coverage score of 0.023 indicates that this particular combination may have provided limited information in identifying the solution. However, the consistency score 0.762 indicates a relatively stable relationship of ~BS and BD across cases before Covid-19. In contrast, there is no direct path in the table that captures the role of BD in managing operating liquidity after COVID-19. This suggests that *BD may have played a minor role in managing operating liquidity before and after Covid-19.*

In Table 5, the path ~BS*~DQ was significant in managing operating liquidity in the fsQCA analysis before and after COVID-19. The negative sign with DQ indicated that it was absent in managing operating liquidity. However, after COVID-19, two paths involving DQ emerged, i.e., OBH*~DQ and WOB*DQ. This new path (WOB*DQ) is significant, as indicated by the raw coverage (0.562), unique coverage (0.059), and consistency (0.829) of this path, suggesting that DQ played a more significant role in managing operating liquidity during the pandemic. Furthermore, *the combination of DQ with WOB was crucial in managing operating liquidity during the crisis*. Therefore, having qualified directors on board, in combination with WOB, plays a pivotal role in managing crises like COVID-19, which supports the glass cliff effect. This study is in line with Ujunwa [91].

## Specific compositions check

Using fsQCA, we derived three potential solutions for the outcome of interest before and after Covid-19. We then evaluated two specific propositions related to WOB and assessed the number of instances in the dataset that support these propositions.[92]. We took two models, WOB*~BS and WOB*DQ, which indicate the number of supporting cases in the dataset to

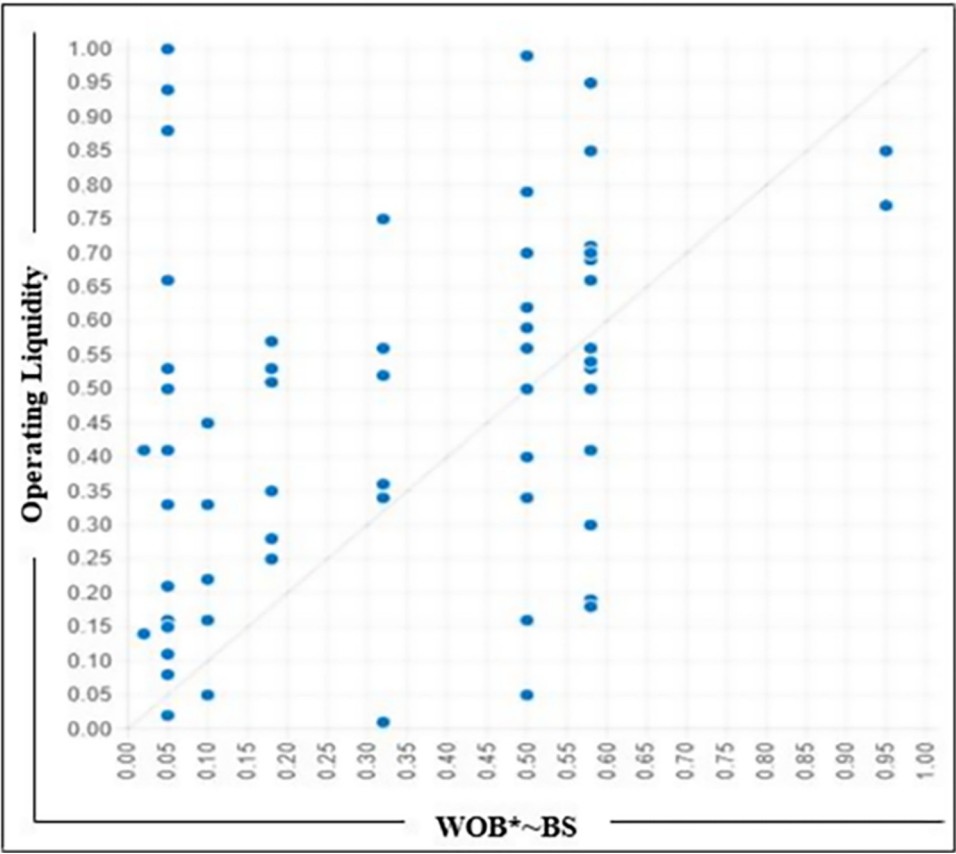

**Fig 2. XY Plot WOB\*~BS.**

depict the relationship between the identified configuration and the outcome of interest. Fig 2 displays the XY plot of the model WOB\*~BS, while Fig 3 exhibits the XY plot of the model WOB\*DQ. Both models partially support the propositions.

## Discussion and conclusion

Using an intermediate number of cases, usually between 15 and 65, fuzzy set qualitative comparative analysis (fsQCA) uses a reciprocal interchange between theories and empirical evidence. The study's objective is to examine the relationship between operational liquidity and corporate governance features while taking the COVID-19 pandemic and the "glass cliff" into consideration. This research examines the corporate environment in 2018 and 2021 in order to evaluate the shifts that occurred both prior to and following the start of the Covid-19 epidemic. fsQCA is used in the analysis to examine the circumstances leading up to and following the pandemic, providing information about changes in the business environment at this time in the context of glass cliff effect. This study purposefully uses important corporate governance input variables, like the number of women on the board (WOB), board size (BS), ownership by blockholders (OBH), director qualifications (DQ), and the number of busy directors (BD), to examine the glass cliff impact. In this analysis, operating liquidity is represented by the current ratio, which acts as the output variable.

The aim of the study is to use fsQCA to do a necessary condition analysis in order to ascertain whether any input variable is necessary for the desired outcome. The use of this analysis is

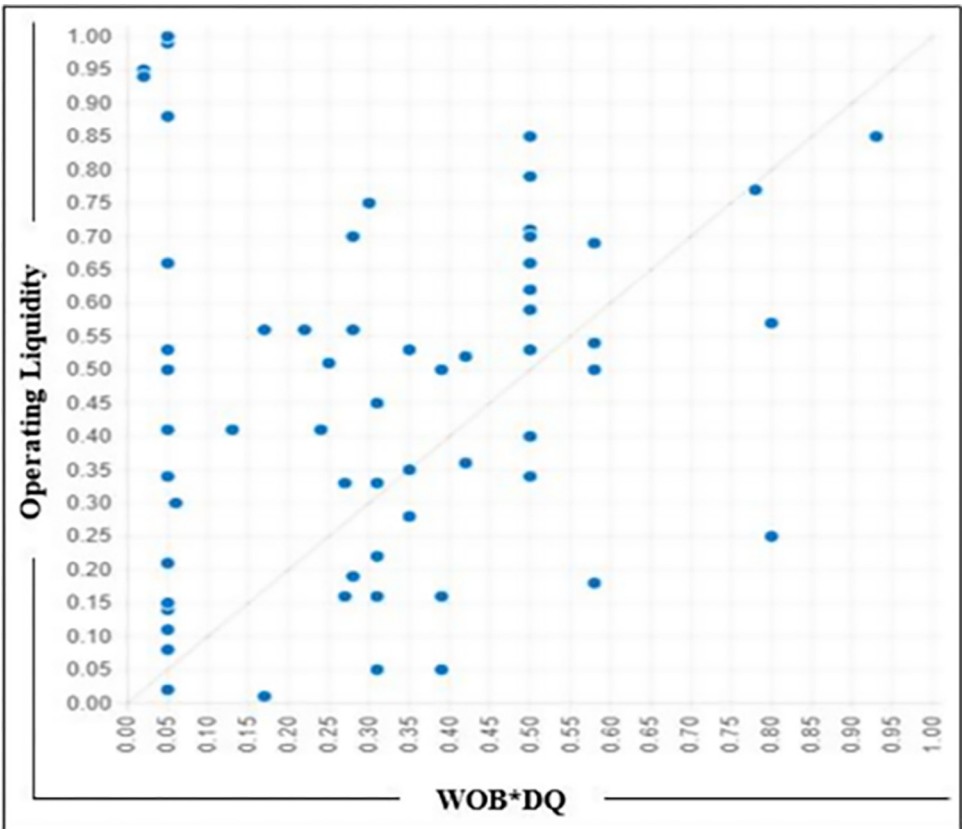

**Fig 3. XY Plot WOB*DQ.**

to pinpoint the circumstances in which a specific variable is required in order to get the desired result. The findings of the necessary condition analysis (NCA) conducted before and after Covid-19 using fuzzy-set qualitative comparative analysis (fsQCA) are shown in Table 2. Overall results indicate, none of the consistency and coverage values simultaneously reach the threshold of 0.90, which is generally regarded as the minimum acceptable level for necessary conditions of individual input variable in fsQCA. This suggests that there are no necessary conditions of any individual input variable before or after Covid-19 for operating liquidity. Therefore, if any of them are not present, operations can be continued. Thus, no input variable like WOB, BS, OBH, DQ, BD is compulsory for the operating liquidity of firms listed on the PSX.

FsQCA multiple solutions provide practical and implementable findings for all levels of management, unlike CRA' single best solution. This makes QCA more aligned with real-life phenomena of human complex system, offering a more comprehensive and practical approach for social science research, social changes and modern theory (Davidsson & Emmenegger, 2013).

The sufficiency analysis results for the model operating liquidity = f (WOB, BS, OBH, DQ, BD) using the Quine-McCluskey method and the most parsimonious specify analysis (fsQCA) are shown in Table 5 for the years 2018 (pre-Covid-19) and 2021 (post-Covid-19). Regression offers a single best solution; in contrast, fsQCA offers several. According to the 2018 analysis, the following three conditions i.e., ~BS*~DQ, ~BS*OBH, and ~BS*BD are sufficient for the outcome. On the other hand, the results for 2021 show that the combinations of WOB*~BS,

OBH*~DQ, and WOB*DQW were found to be sufficient for the desired objective, indicating that the presence of women in corporate leadership roles is associated with increased operating liquidity. These findings align with the concept of the "glass cliff," which posits that women are more likely to be placed in leadership positions during times of crisis or within struggling businesses [86]. Consequently, the study validates the presence of the glass cliff effect during the COVID-19 pandemic, emphasizing that "*an increased representation of women on corporate boards is expected to play a crucial role in effectively managing liquidity, particularly amid the challenges posed by the COVID-19 pandemic*."

However, the study did not find evidence to suggest that a larger board size plays a role in managing firm liquidity in the Covid-19 pandemic. Our study is inline with the study of [50]. Moreover, the study found that OBH played a significant role in operating liquidity before and after the Covid-19 pandemic, but the specific paths that captured this relationship differed. While ~BS*OBH was a critical path before Covid-19, OBH*~DQ emerged as the more important path after Covid-19. Therefore, it can be affirmed that the ownership of a blockholder plays a crucial role in corporate governance, influencing the control of operational liquidity both before and after the onset of the COVID-19 pandemic. It is notewothy to mention here that The path also highlights the importance of women in managing operating liquidity, especially in the context of the Covid-19 crisis, in combination with the director's qualification-*the combination of director's qualifications and WOB emerges as highly significant for effective crisis management*. Therefore, by examining whether board characteristics—in particular, the representation of women on the board—significantly influence a firm's operational liquidity, especially in light of the glass cliff effect, the study effectively achieves its objectives.

The study reveals important and unexpected facts that greatly add to the corpus of existing research on corporate governance in folling ways: First, the essential condition analysis (NCA) emphasises that firms can operate without reliance on any particular variable taken in the study. This suggests that the continued operations of businesses are not hampered by the lack of a certain variable. Second, the sufficiency analysis provided an expanded understanding of the conditions that are adequate for the operational liquidity model both before and during the Covid-19 period through the application of the Quine-McCluskey technique and fsQCA. The analysis conducted for 2021 indicates a relationship between increased operational liquidity and the representation of women in business leadership positions. This aligns with the "glass cliff" theory, emphasising the increased presence of women in positions of leadership during emergencies. Third, the research could not uncover any evidence to support the proposition that managing business liquidity during the Covid-19 epidemic is influenced by a higher board size. However, this result goes against some previous research. Fourth, the study underscores the crucial role of blockholders ownership in corporate governance, influencing operational liquidity control liquidity both before and after the Covid-19 pandemic. Fifth, the study emphasised how having more women on boards (WOB) and having qualified directors (DQ) can work together to manage crises effectively, as demonstrated during the Covid-19 pandemic. This emphasises how crucial it is to have a good composition of BODs and competent leadership team when managing operating liquidity during difficult circumstances.

## Future research directions

a. The study may not fully represent the non-financial sector as a whole because it only includes data from 60 non-financial firms that are listed on the Pakistan Stock Exchange's (PSX) KSE-100 index. Furthermore, in order to validate more broadly applicable findings in future studies, it might be essential to increase the sample size to incorporate information from other nations in the region like South Asia.

b. The study only analyzes data from 2018 to 2021, which may not provide a comprehensive understanding of the effects of COVID-19 on non-financial companies in Pakistan. Future studies could expand the timeframe to provide a more complete picture.

c. The study only examines six variables (WOB, BS, OBH, DQ, BD, and CR), which may only capture some relevant condition that impact the performance in terms of operating liquidity of non-financial companies in Pakistan. Future studies could consider including additional variables to provide a more comprehensive analysis.

d. The study only examines the asymmetric relationships between variables but does not establish symmetric causality. Therefore, other factors may be impacting the results. Future studies could use experimental or longitudinal designs to establish causality.

## Supporting information

**S1 Data.**
(PDF)

## Author Contributions

**Conceptualization:** Syed Maisam Raza Rizvi.

**Data curation:** Dongli Cao, Syed Maisam Raza Rizvi.

**Formal analysis:** Dongli Cao, Safdar Husain Tahir.

**Funding acquisition:** Dongli Cao, Khuda Bakhsh Khan.

**Investigation:** Dongli Cao.

**Methodology:** Safdar Husain Tahir, Syed Maisam Raza Rizvi.

**Project administration:** Khuda Bakhsh Khan.

**Resources:** Dongli Cao, Khuda Bakhsh Khan.

**Software:** Dongli Cao, Safdar Husain Tahir, Syed Maisam Raza Rizvi.

**Validation:** Syed Maisam Raza Rizvi.

**Visualization:** Khuda Bakhsh Khan.

**Writing – original draft:** Safdar Husain Tahir, Syed Maisam Raza Rizvi.

**Writing – review & editing:** Dongli Cao, Safdar Husain Tahir, Syed Maisam Raza Rizvi.

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
