## [Decision Letter · Decision Letter 0]

2 Jan 2024

PONE-D-23-36238Exploring the Influence of Women's Leadership and Corporate Governance on Operational Liquidity: The Glass Cliff EffectPLOS ONE

Dear Dr. Tahir,

Thank you for submitting your manuscript to PLOS ONE. After careful consideration, we feel that it has merit but does not fully meet PLOS ONE’s publication criteria as it currently stands. Therefore, we invite you to submit a revised version of the manuscript that addresses the points raised during the review process.

We look forward to receiving your revised manuscript.

Kind regards,

Syed Usman Qadri, PhD

Academic Editor

PLOS ONE

Journal Requirements:

Additional Editor Comments:

**ACADEMIC EDITOR: Reviewers have now commented on your paper. You will see that there are a number of issues that need to be addressed before the paper can be accepted for publication.**

Reviewers' comments:

Reviewer's Responses to Questions

**Comments to the Author**

1. Is the manuscript technically sound, and do the data support the conclusions?

Reviewer #1: Partly

Reviewer #2: Partly

Reviewer #3: Yes

2. Has the statistical analysis been performed appropriately and rigorously? 

Reviewer #1: Yes

Reviewer #2: Yes

Reviewer #3: Yes

3. Have the authors made all data underlying the findings in their manuscript fully available?

Reviewer #1: Yes

Reviewer #2: Yes

Reviewer #3: Yes

4. Is the manuscript presented in an intelligible fashion and written in standard English?

Reviewer #1: No

Reviewer #2: No

Reviewer #3: Yes

5. Review Comments to the Author

Reviewer #1: 1.What motivated exploring the influence of women’s leadership and corporate governance on operational liquidity within the context of the Glass Cliff effect?

2.Can you elaborate on why the Glass Cliff concept is particularly relevant to studying operational liquidity in the context of women’s leadership and corporate governance?

3.How specifically do the research objectives aim to address or contribute to our understanding of the Glass Cliff effect in the context of operational liquidity?

4.Is there a specific theoretical framework that underpins the study, and how does it guide the exploration of the research objectives?

5.The literature review provided in the article needs to establish a robust foundation to support the formulated hypothesis. While the introduction presents clear research objectives and questions related to the influence of women's leadership and corporate governance on operational liquidity, the literature review needs to adequately synthesize existing knowledge and evidence in a manner that logically leads to the proposed hypothesis. To enhance the quality of the literature review, it is recommended that the authors revisit and strengthen the synthesis of existing literature, emphasizing studies that directly contribute to the understanding of the Glass Cliff effect and its potential implications for operational liquidity. Also, the literature review should clearly show how the evidence being looked at led to the development of the hypothesis. This ensures a clear and logical flow from the existing body of knowledge to the specific research question being looked at. Addressing this gap will not only strengthen the study's theoretical foundation but also enhance its overall contribution to the field.

6.Some important relevant papers are missing in the literature review, for instance:

A. Exploring the China-Pakistan economic corridor project performance during Covid-19 pandemic

B. Effect of firm structure on corporate cash holding (evidence from non-financial companies).

7. The choice to utilize data spanning the years 2018 to 2021 is noted. However, considering the dynamic nature of business environments and the potential impact of external factors, it would be beneficial to understand the rationale behind not incorporating more recent data in the analysis. Recent years have witnessed significant global events, economic shifts, and changes in corporate landscapes that could influence the variables under investigation. Including more up-to-date information could provide a more accurate reflection of the current state of affairs and enhance the relevance and applicability of the study’s findings. The authors should address this concern by justifying the selected timeframe or, if feasible, incorporating more recent data to ensure the study’s results reflect the latest trends and developments in the field.

8. The conclusion should go beyond summarizing the findings and reiterate the study's broader implications. It would be beneficial for the authors to contextualize the significance of their results within the existing literature, restate the critical contributions of the research, and discuss any unexpected or noteworthy findings. Additionally, the conclusion should tie back to the research objectives and emphasize how the study addresses the initial research questions.

9. While the current study makes a valuable contribution to understanding the influence of women's leadership and corporate governance on operational liquidity, it is essential to note that the article lacks a discussion on future research directions and policy implications. Addressing these aspects would significantly enhance the impact and practical relevance of the research.

10. Consider incorporating a brief section that outlines potential avenues for future research. This could include suggestions for further exploring specific aspects of the Glass Cliff effect, operational liquidity, or related variables not covered in the current study.

Reviewer #2: Introduction

•Can the author(s) enhance the motivation of their study even more?

•Additionally, the author(s) must present the sub-research questions and objectives in a concise and organized manner using bullet points.

Literature Review

•The mere provision of the framework in section 2.6 by the author(s) is insufficient. Therefore, it is recommended that the author(s) include a detailed description of the proposed conceptual framework in the study.

Methodology

•A proposed renaming of Section 3.0 to "Method" is desired.

•It is recommended that the authors substitute "relationship" for "connection."

•Figure 2 ought to be redrawn by the authors.

Discussion and Conclusion

•The present discussion and concluding remarks fail to provide a clear depiction of the subject matter that the author intended to investigate. The following is the recommended order for the authors to rewrite this section:

1.The discussion section should be redrafted to explain extensively the implications of the findings. This should be done in line with previous studies.

2.Additionally, the implications of the study must be specified in detail.

3.The conclusion should also be redrafted ensuring that they restate the topic and its significance. In addition, the author(s) should restate the study's claim. Address opposing viewpoints and explain why the readers should support their viewpoint. Include a call to action or a summary of the possibilities and limitations of future research.

Reviewer #3: I suggest to do more realistic analysis considering other corporate sectors too in other countries as well. Analysis outcome may differ. I suggest to enhance variable counts to be closer to realistic corporate environment.

6. PLOS authors have the option to publish the peer review history of their article (what does this mean?). If published, this will include your full peer review and any attached files.

Reviewer #1: No

Reviewer #2: No

Reviewer #3: No

---

## [Author Response · Author response to Decision Letter 0]

16 Feb 2024

The article has been revised in accordance with the feedback provided by Reviewer-1, Reviewer-2, and Reviewer-3.Please find attached a clean copy of revised version of the article, accompanied by a revised copy wherein changes have been highlighted.

---

## [Decision Letter · Decision Letter 1]

1 Apr 2024

Exploring the Influence of Women's Leadership and Corporate Governance on Operational Liquidity: The Glass Cliff Effect

PONE-D-23-36238R1

Dear Dr. Tahir,

We’re pleased to inform you that your manuscript has been judged scientifically suitable for publication and will be formally accepted for publication once it meets all outstanding technical requirements.

Kind regards,

Syed Usman Qadri, PhD

Academic Editor

PLOS ONE

Additional Editor Comments (optional):

Reviewers' comments:

Reviewer's Responses to Questions

**Comments to the Author**

1. If the authors have adequately addressed your comments raised in a previous round of review and you feel that this manuscript is now acceptable for publication, you may indicate that here to bypass the “Comments to the Author” section, enter your conflict of interest statement in the “Confidential to Editor” section, and submit your "Accept" recommendation.

Reviewer #1: All comments have been addressed

Reviewer #2: All comments have been addressed

2. Is the manuscript technically sound, and do the data support the conclusions?

Reviewer #1: Yes

Reviewer #2: Yes

3. Has the statistical analysis been performed appropriately and rigorously? 

Reviewer #1: Yes

Reviewer #2: Yes

4. Have the authors made all data underlying the findings in their manuscript fully available?

Reviewer #1: Yes

Reviewer #2: Yes

5. Is the manuscript presented in an intelligible fashion and written in standard English?

Reviewer #1: Yes

Reviewer #2: Yes

6. Review Comments to the Author

Reviewer #1: I am please to inform you that all of the comments and suggestions raised during the review process have been addressed. I am happy to announce my decision to accept the paper for publication.

Reviewer #2: I am grateful that you have taken the time to address all of the concerns that I raised concerning your article. The readership will be provided with knowledge that is both perceptive and interesting, as this study is both engaging and informative.

7. PLOS authors have the option to publish the peer review history of their article (what does this mean?). If published, this will include your full peer review and any attached files.

Reviewer #1: No

Reviewer #2: No

---

## [Editor Report · Acceptance letter]

14 May 2024

PONE-D-23-36238R1 

PLOS ONE

Dear Dr. Tahir, 

I'm pleased to inform you that your manuscript has been deemed suitable for publication in PLOS ONE. Congratulations! Your manuscript is now being handed over to our production team.

Kind regards, 

on behalf of

Dr. Syed Usman Qadri 

Academic Editor

PLOS ONE